# Decreased Oligodendrocyte Number in Hippocampal Subfield CA4 in Schizophrenia: A Replication Study

**DOI:** 10.3390/cells11203242

**Published:** 2022-10-15

**Authors:** Andrea Schmitt, Laura Tatsch, Alisa Vollhardt, Thomas Schneider-Axmann, Florian J. Raabe, Lukas Roell, Helmut Heinsen, Patrick R. Hof, Peter Falkai, Christoph Schmitz

**Affiliations:** 1Department of Psychiatry and Psychotherapy, University Hospital, Ludwig Maximilian University (LMU) Munich, 80336 Munich, Germany; 2Laboratory of Neuroscience (LIM27), Institute of Psychiatry, University of São Paulo, São Paulo 05403-903, Brazil; 3Institute of Anatomy, Faculty of Medicine, Ludwig Maximilian University (LMU) Munich, 80336 Munich, Germany; 4Morphological Brain Research Unit, Department of Psychiatry, University of Würzburg, 97080 Würzburg, Germany; 5Institute of Forensic Pathology, University of Würzburg, 97078 Würzburg, Germany; 6Nash Family Department of Neuroscience, Icahn School of Medicine at Mount Sinai, New York, NY 10029-6574, USA; 7Friedman Brain Institute, Icahn School of Medicine at Mount Sinai, New York, NY 10029-6574, USA; 8Max Planck Institute of Psychiatry, 80804 Munich, Germany

**Keywords:** schizophrenia, hippocampus, CA4, dentate gyrus, postmortem, stereology, oligodendrocyte, neuron

## Abstract

Hippocampus-related cognitive deficits in working and verbal memory are frequent in schizophrenia, and hippocampal volume loss, particularly in the cornu ammonis (CA) subregions, was shown by magnetic resonance imaging studies. However, the underlying cellular alterations remain elusive. By using unbiased design-based stereology, we reported a reduction in oligodendrocyte number in CA4 in schizophrenia and of granular neurons in the dentate gyrus (DG). Here, we aimed to replicate these findings in an independent sample. We used a stereological approach to investigate the numbers and densities of neurons, oligodendrocytes, and astrocytes in CA4 and of granular neurons in the DG of left and right hemispheres in 11 brains from men with schizophrenia and 11 brains from age- and sex-matched healthy controls. In schizophrenia, a decreased number and density of oligodendrocytes was detected in the left and right CA4, whereas mean volumes of CA4 and the DG and the numbers and density of neurons, astrocytes, and granular neurons were not different in patients and controls, even after adjustment of variables because of positive correlations with postmortem interval and age. Our results replicate the previously described decrease in oligodendrocytes bilaterally in CA4 in schizophrenia and point to a deficit in oligodendrocyte maturation or a loss of mature oligodendrocytes. These changes result in impaired myelination and neuronal decoupling, both of which are linked to altered functional connectivity and subsequent cognitive dysfunction in schizophrenia.

## 1. Introduction

Abnormalities in the cortical cytoarchitecture in dementia praecox, later termed schizophrenia, were first reported by Alzheimer who described “damage to or destruction of the nervous cortical elements” [1]. However, the unreplicated qualitative findings of neuronal abnormalities and lack of quantitative methods made it difficult to uncover a histological substrate of schizophrenia, leading to the conclusion that schizophrenia has no neuropathological basis [2] and that the search for the causes should focus on functional abnormalities. However, pneumencephalographic and early brain imaging studies revealed general morphological abnormalities, such as ventricular enlargement [3]; these studies were followed by magnetic resonance imaging (MRI) evaluations, which showed more subtle brain atrophy and alterations in frontotemporal connectivity [4]. Of these findings, hippocampal volume reduction has been among the most frequently replicated ones in schizophrenia research [5].

The hippocampus plays a role in working memory and visual and verbal learning, and deficits in these cognitive domains, along with reduced volumes and altered connectivity, have consistently been described in clinical and MRI studies in patients with first- and multi-episode schizophrenia [6,7,8,9]. Patients with schizophrenia show greater reductions in CA2/3 and CA4/dentate gyrus (DG) subfield volume than individuals in the prodromal period of psychosis, who in turn have significantly greater volume reductions in these subfields than healthy volunteers. In schizophrenia, these subregional volumes were associated with cognitive deficits related to visual, verbal, and working memory [5]. A meta-analysis of studies in first-episode schizophrenia found decreased fractional anisotropy in the circuitry of frontal-limbic regions, indicating disturbed myelination and connectivity of the hippocampus [4]. However, the underlying cellular alterations remain elusive and require postmortem investigations.

Earlier studies focusing on the hippocampus in schizophrenia described smaller volumes and fewer pyramidal neurons in the CA but did not find changes in granular neurons in the DG [10,11]. A recent meta-analysis on subfield volumes and neuron number, density, and size reported reductions in the volumes of all subregions, including CA4, and in the number of neurons in the left hippocampal CA1, CA2/3, and subiculum regions but found no significant differences in neuron density [12]. Cell density studies have a methodological bias that influences volume; this bias is caused by shrinkage of tissue during fixation and staining or neuropil loss, irregularities in the shape and size of cells, fixed cell orientation, and sectioning-related cell damage [13]. Design-based stereology studies overcome this bias, but their results show no significant decrease in neurons in CA2/3 [12]. In a study in patients with schizophrenia and controls, we used unbiased stereology to investigate the numbers of neurons and glial cells in CA1, CA2/3, and CA4 and the subiculum and DG of the posterior hippocampus and found no changes in neuron or astrocyte numbers in the CA fields or subiculum [14]. Unexpectedly, we detected a bilateral reduction in oligodendrocyte number specifically in CA4, also referred to as the hilar part of CA3 [15] (pp. 37–115). In the left hemisphere, this reduction was also observed in the anterior hippocampus, and the number of neurons in the DG was reduced [16]. When we examined the whole hippocampus, we found that the left CA4 subregion was significantly smaller and contained significantly fewer oligodendrocytes compared to controls [16]. However, the results need to be replicated in a study with larger sample size.

The present postmortem study used rigorous, design-based stereology to assess neuron, oligodendrocyte, and astrocyte numbers and density in the right and left CA4 and DG of the brains of patients with schizophrenia and matched controls. In addition, it measured the volumes of CA4 and the granular cell layer of the DG in these regions of interest (Figure 1).

The investigated brains were the same as those analyzed in previous stereology studies by our group [17,18,19,20]. With this approach, we also found decreased numbers and a lower density of oligodendrocytes in the right and left CA4; however, we found no significant intergroup differences in the number and density of neurons or astrocytes or the volumes of subregions.

## 2. Materials and Methods

### 2.1. Brains

The brain autopsy tissue evaluated in this study was provided by the Heinsen Collection (University of Wuerzburg, Wuerzburg, Germany). The study was approved by the ethics committee of the University of Wuerzburg. Brains from 11 men with schizophrenia (mean age ± standard error of the mean [SEM], 50.9 ± 3.7 years; mean postmortem interval [PMI; time between death and autopsy], 39.4 ± 8.3 h; and mean fixation time, 187.5 ± 24.5 days) and 11 age-matched male controls (mean age, 54.5 ± 7.1 years; mean PMI, 23.5 ± 4.3 h; and mean fixation time, 1028.3 ± 432 days) that included the whole hippocampus were available for analysis (Table 1).

Patients fulfilled the diagnostic criteria for schizophrenia in the Diagnostic Statistical Manual, 4th revision (DSM-IV), and International Statistical Classification of Diseases and Related Health Problems, 10th revision (ICD-10) and had been treated in German clinical facilities (state psychiatric hospitals and local district hospitals). All of them had been treated with antipsychotics during most of their illness and medical records were available for all cases, but the last medication doses and lifetime medication exposures were not available. Two experienced psychiatrists assessed the reports to ensure the absence of psychiatric diagnoses in the controls and to verify that the diagnoses of schizophrenia complied with DSM-IV and ICD-10 criteria. In addition, they reviewed the autopsy records, including a brief medical history. The ethnic background of patients with schizophrenia and controls was white European, but the groups were not matched with respect to socioeconomic status or education because the patients had lower education status. In both groups, exclusion criteria included neurological diseases requiring treatment or that may have influenced cognitive tests (e.g., stroke with aphasia); history of recurrent seizures lifetime severe craniocerebral trauma with loss of consciousness; history of alcohol or substance abuse disorders; and type 2 diabetes with a free plasma glucose level greater than 200 mg/dL [18]. However, suicide-related head trauma cannot be excluded. All brain sections were controlled for gross neuropathological or morphological changes.

### 2.2. Tissue Processing

All tissues were fixed and processed in the same way at the Morphological Brain Research Unit (University of Wuerzburg, Wuerzburg, Germany); all brains were embedded in gelation except C7, for which celloidin was used. For detailed information, see Kreczmanski et al. [17], who investigated the same postmortem brains in previous studies. Briefly, for fixation, brains were immersed for 4 weeks or more in 10% formalin (1 part commercial 40% aqueous formaldehyde plus 9 parts tap water). Then, the brainstem and cerebellum were separated at the rostral pons and a medio-sagittal incision was made to separate the hemispheres. Subsequently, the hemispheres were placed for 8 days in a cryoprotective solution consisting of formaldehyde, dimethyl sulfoxide, and glycerol, embedded as described above, and frozen in isopentane at −60 °C. The deeply frozen blocks were then serially cut into sections of 600 to 700 µm with a special sliding microtome (Jung, Nußloch, Germany) [21]. The serial slices were stored in 4% formalin. Gallocyanin (a Nissl stain) was used to stain every second or third (see [21]). Brain C7 was dehydrated in a graded series of ethanol solutions and mounted in celloidin. The hemispheres were dissected into 440-µm thick sections with a sliding microtome (Polycut, Cambridge Instruments, London, UK) [22]. The mean number of whole-brain sections used were as follows: controls, 12 ± 2.17; patients, 10 ± 2.34.

The whole hippocampus was analyzed, with a focus on the CA4 and the DG (Figure 1). We did not analyze the anterior and posterior hippocampus separately because separating the two parts by following external landmarks such as the lateral geniculate nucleus may be a source of possible bias [14]. In CA4, we used histological and morphological criteria based on the histological staining at 40× magnification to differentiate between neurons; small, dark cells (oligodendrocytes); and large, pale cells (astrocytes) (Figure 2 and Figure 3).

Typical findings in neurons were a large cytoplasm, a less distinct nuclear membrane, a pale nucleus in a clearly visible nucleolus, and a more uniform distribution of nuclear chromatin material. Mature oligodendroglia showed no staining in the cytoplasm, no nucleolus, and pronounced nuclear staining with scattered chromatin. We also included immature oligodendrocytes, which are larger and paler than mature ones [23]. Astrocyte staining was less dense [24,25,26]. In the DG, we focused on the densities and numbers of granule cells (Figure 3).

### 2.3. Stereological Analyses

Stereological analyses used a stereological workstation comprising a light microscope (Zeiss Axio Imager M2^®^, Carl Zeiss Microscopy GmbH, Jena, Germany); Zeiss objectives (1.25×, 2.5×, 5×, 10×, 20×, 40×, and 100×); oil; a stage controller; software (Stereo Investigator^®^ 2018.2.2 64 bit, MicroBrightField Bioscience Williston, VT, USA); and a 27.2-in monitor. The CA4 and DG hippocampal subregions (Figure 1) were delineated at a magnification of 1.25x from the anterior portion to the corpus callosum splenium. Cavalieri’s principle was used for calculating overall volumes [27], and the optical fractionator method, for estimating cell numbers [27,28,29]. For neurons, astrocytes, and oligodendrocytes, we used the prediction methods described by Schmitz [28] and Schmitz and Hof [29] to calculate the predicted coefficient of error in the estimates of total numbers. The rater was blind to diagnosis. The method used for the stereological counting of cells is presented in Table 2.

### 2.4. Statistical Analyses

Findings in patients with schizophrenia and healthy controls were compared with a general linear model, i.e., a multivariate analysis of covariance (MANCOVA), with diagnosis as the between-subject factor, hemisphere as the within-subject factor, and age and PMI as covariates.

Because of differences in the distribution of fixation time and subsequent deviation from the normal distribution, non-parametric Spearman correlations were calculated for the influence variables fixation time, PMI, and age. The fixation time had no significant effect on the investigated variables, so it was not used in the MANCOVA. In case of significant results in the MANCOVA, post hoc tests were performed as univariate analyses of covariance separately for the left and right sides. A *p* value of less than 0.005 or less than 0.025 for post hoc analyses (Bonferroni corrected for 2 variables, *left* and *right*) was considered to indicate a statistically significant difference (after Bonferroni correction for 10 variables). SPSS version 28 (IBM, Armonk, NY, USA) was used for statistical analyses.

## 3. Results

The mean total number of oligodendrocytes (small, dark cells) in CA4 was significantly lower in patients than in controls (left hemisphere, −28%; right hemisphere, −39%; F (1, 18) = 13.36, *p* = 0.002; Figure 4J). Neither the patients nor the controls showed significant differences between the two hemispheres (F(1, 18) = 0.73, *p* = 0.40). Because of significant correlations (see below), results were adjusted by the covariates age and PMI. The post hoc univariate analyses of covariance performed for each hemisphere separately confirmed that, compared with controls, schizophrenia patients had significantly fewer oligodendrocytes in CA4 in both the left (F1, 18) = 11.47, *p* = 0.003) and the right hippocampus (F(1, 18) = 11.09, *p* = 0.004).

Notably, the mean total number (Figure 4J) and mean density of oligodendrocytes in CA4 were significantly lower in patients than in controls (left, −17%; right, −28%; F(1, 18) = 17.03, *p* < 0.001; Figure 5J). The hemispheres showed no significant differences in either group (F(1, 18) = 0.91, *p* = 0.35). The post hoc analyses performed for each hemisphere separately confirmed a significantly lower oligodendrocyte density in the left (F(1, 18) = 8.69, *p* = 0.009) and right hemisphere (F(1, 18) = 16.24, *p* = 0.001). We did not find differences in oligodendrocyte number or density between suicide victims and patients with a non-suicidal cause of death. The mean volumes of the subregions (Figure 6A,D), mean number of neurons (Figure 4A,D), and mean densities of neurons (Figure 5A,D) in CA4 and the DG were similar in both groups. In addition, there were no indications of astrocytosis in either group, i.e., no differences in number and density of astrocytes (large, pale cells) were seen (Figure 4G and Figure 5G).

In controls, the left DG volume and PMI were significantly correlated (ρ = 0.62, *p* = 0.042). In patients, age had a significant effect on mean astrocyte number in the left CA4 (ρ = −0.73, *p* = 0.011), mean oligodendrocyte number in the right CA4 (ρ = −0.71, *p* = 0.015), and mean oligodendrocyte density in the right CA4 (ρ = −0.65, *p* = 0.031). Therefore, the MANCOVA results were adjusted for age and PMI. No correlations with the duration of fixation were detected. The results from brain C7 (which was embedded in celloidin) did not systematically deviate from those in the other gelatin-embedded brains.

## 4. Discussion

By using design-based stereology, we detected a decreased number and density of oligodendrocytes in CA4 in both hemispheres in schizophrenia and thus replicated our previous results in the posterior hippocampus [14] and expanded our findings from the anterior and whole hippocampus [16]. In contrast to our stereological study on a different brain collection [16], we found no significant differences between hemispheres in patients with schizophrenia and controls, suggesting an absence of significant hemispheric asymmetries. Our studies clearly identify the role of oligodendrocytes in hippocampal subfields in schizophrenia. Another stereological study that evaluated oligodendrocytes, astrocytes, and microglia reported lower total numbers of neurons and glia in CA4 and other subregions of the hippocampus in patients with depression and schizophrenia than in controls [30]. However, the present study did not detect differences in astrocyte and neuron numbers or densities. A recent meta-analysis of postmortem hippocampal subfields in schizophrenia that focused only on neuronal number, density, and size found no changes in neuron density; when only studies that used stereological techniques were evaluated, neuron number was reduced only in the left CA1 and left subiculum [12].

The present study found no reduction in CA4 or DG volumes; however, these findings do not agree with the results of the above-mentioned meta-analysis, in which 10 studies showed a volume reduction of the left CA4 and 7 studies, a reduced volume of left DG [12]. Another previous study found smaller volumes of the CA4 and DG in the left anterior hippocampus that correlated with the numbers of oligodendrocytes and neurons, respectively. Previously, we detected a reduced volume of the left CA4 [16].

We investigated only male patients, so we do not know whether our results apply also to female patients. In our previous research on the posterior hippocampus, CA4 volume and neuron number were significantly smaller in women than in men [14]. In the anterior hippocampus, we found sex-related effects also in neuron and astrocyte numbers in the right CA4 and in the volume of this subregion, all of which were smaller in women [16]. However, in line with previous studies [14,16], the present study did not find astrocytosis in schizophrenia cases, confirming that schizophrenia is not a classical neurodegenerative disease with increased astrocyte numbers and neuronal loss.

Our study used design-based stereology to evaluate Gallocyanin-stained histological sections and identified oligodendrocytes only on the basis of morphological criteria. An immunohistochemical assessment found no significant differences in the density of oligodendrocytes expressing oligodendrocyte transcription factor 1 (Olig1) or 2 (Olig2) between patients with schizophrenia and controls in the subregions of the posterior hippocampus [31]. However, these specific Olig1 and Olig2 antibodies stain only subpopulations of oligodendrocytes at overlapping maturation stages, and to date, it is unclear whether oligodendrocyte precursor cells fail to differentiate properly into mature oligodendrocytes or whether mature oligodendrocytes degenerate or undergo apoptosis. In gene expression studies, downregulation of proteins related to oligodendrocytes has been repeatedly observed in the hippocampus of patients with schizophrenia [32,33], suggesting dysfunctional oligodendrocytes. Moreover, a stereology study of the superior frontal cortex detected a decreased oligodendrocyte number in schizophrenia [24]. Therefore, this finding may not be region specific, although another study did not detect decreased oligodendrocyte numbers in the anterior cingulate cortex [34].

Oligodendrocytes play a role in the myelination of neurons and neuronal network connectivity [35]. In addition to their role in myelination and nerve impulse propagation, they supply energy to neuronal axons [36]. Besides myelination of projecting axons of pyramidal neurons [37], recent studies revealed myelination of parvalbumin-expressing interneurons [38]. Disturbed cross-talk between oligodendrocytes and interneurons may lead to improper local connectivity in schizophrenia [39]. Diffusion tensor imaging (DTI) studies found deficits in myelination and reduced fractional anisotropy as a marker of myelination in the hippocampus and fornix of patients with schizophrenia [40,41]; these changes were associated with cognitive deficits [42]. White matter fractional anisotropy values were significantly lower in patients with schizophrenia and cognitive deficits than in those with high cognitive performance [43]. A DTI study found a relationship between variants of genes related to oligodendrocytes, e.g., myelin-associated glycoprotein, and the integrity of white matter tracts and cognition in patients with schizophrenia [44]. We used data from our previous stereological study in the hippocampus and medical records from the patient cohort to classify cognitive deficits and showed significantly lower oligodendrocyte numbers in patients with definite cognitive deficits than in those with only possible cognitive deficits [31].

Our study did apply morphological criteria, so we were not able to analyze pyramidal neurons and interneurons separately [14]. A stereological study with immunohistochemical staining of interneurons in schizophrenia found lower numbers of somatostatin-expressing interneurons in CA4, CA2/3, and CA1 and of parvalbumin-immunoreactive interneurons in CA4 and CA1 but no changes in total neuron number [45]. Postmortem studies have repeatedly shown reduced expression of parvalbumin in schizophrenia [46,47,48] along with increased methylation of parvalbumin in the hippocampus [49].

We identified cell types by histology, but we did not confirm results by immunohistochemical staining of oligodendrocytes. The latter is not possible on thick sections of the human brain as used in the present study, as antibodies usually do not penetrate through thick tissue sections [50]. Measurements of oligodendrocyte densities at the surface of thick sections of the human brain processed with immunohistochemistry would neither have been an option. This is due to the fact that cell density measurements may be biased because of shrinkage of the tissue after fixation and staining procedures [29,50]. Furthermore, cell density measurements can in principle not serve as a surrogate of cell number measurements, because cell densities can change without a corresponding change in the number of the same cells [29]. Conversely, cell numbers can change without a corresponding change in the density of the same cells [51]. This is in line with the following finding. We first performed design-based stereological investigations of different cell types on 40-µm thick, histologically stained, serial, human whole-brain sections from the Magdeburg brain collection and showed a loss of oligodendrocytes in the CA4 region [14,16]. We then assessed the density of oligodendrocytes in adjacent sections labeled with Olig1 and Olig2 antibodies [31], and only found a non-significant trend for correlation between histologically identified oligodendrocyte density and Olig2 immunolabeling-based oligodendrocyte density in 10 patients with schizophrenia (*r* = 0.552, *p* = 0.098; unpublished results).

It should be noted that the present sample was small because of the limited availability of serially prepared postmortem brain sections from patients with schizophrenia and low autopsy rates. However, the study was independent and replicated previous findings. In addition, all patients were treated with antipsychotics, so the effects of these drugs cannot be ruled out. However, a stereological study that examined the parietal cortex of monkeys after haloperidol or olanzapine treatment showed no significant reduction of oligodendrocytes immunostained with CNPase [52]. In the mouse hippocampus, haloperidol treatment even increased the number of Olig2-expressing oligodendrocytes and activated quiescent oligodendrocytes [53]. Future stereological studies in animal models with impaired oligodendrocyte function are warranted to investigate the effects of antipsychotic treatment. The PMI may affect volumes of hippocampal subregions because we found that the volume of the left DG was positively correlated with PMI in controls. In patients with schizophrenia, age was negatively correlated with the mean astrocyte number in the left CA4, mean oligodendrocyte number in the right CA4, and mean oligodendrocyte density in the right CA4. Therefore, the effects of age cannot be excluded; however, our results were adjusted for age and PMI and patients and controls were matched for age. As the last point, we did not perform neuropathological staging to exclude the presence of neurodegenerative changes in the brains, but we excluded individuals with neurological diseases.

In summary, we replicated previous findings of lower numbers and density of oligodendrocytes in CA4 in men with schizophrenia. In contrast, we did not find reduced volumes or numbers of neurons in the CA4 and DG subregions. These results may be related to the effects of sex and the small sample size. A decreased oligodendrocyte number indicates a deficit in the differentiation of oligodendrocyte precursor cells, which warrants more detailed evaluation in immunohistochemical postmortem studies of the CA4 with markers of oligodendrocyte precursors at the early myelination stage [54]. Such studies may result in new treatments that target oligodendrocyte precursor cells, which are capable of remyelination [55], to overcome impaired connectivity in schizophrenia.

## Figures and Tables

**Figure 1 cells-11-03242-f001:**
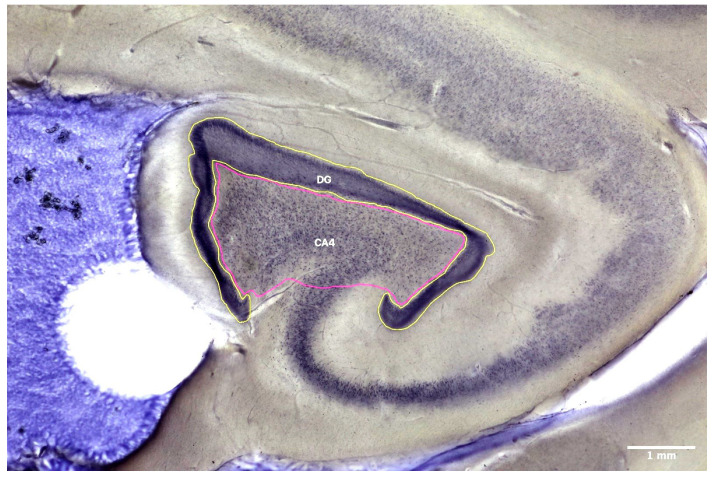
Representative section from the hippocampus of a patient with schizophrenia (1.25× magnification). The cornu ammonis 4 (CA4) is shown in purple, and the dentate gyrus (DG) in yellow.

**Figure 2 cells-11-03242-f002:**
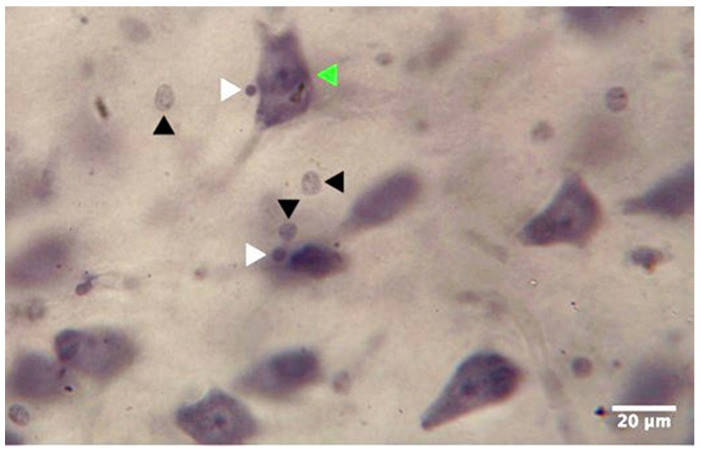
Photomicrograph from cornu ammonis 4 at 40× magnification showing neurons (green arrowhead), astrocytes (black arrowheads), and oligodendrocytes (white arrowheads).

**Figure 3 cells-11-03242-f003:**
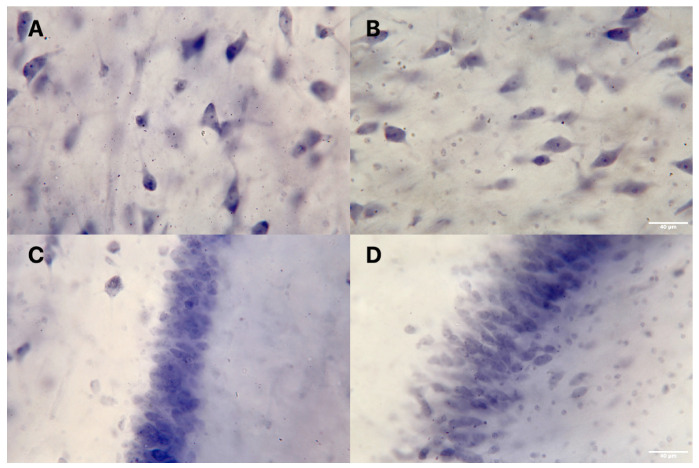
Representative photomicrographs of the cornu ammonis 4 (40× magnification) from a patient with schizophrenia (**A**) and a healthy control (**B**) and of the dentate gyrus from a patient with schizophrenia (**C**) and a healthy control (**D**).

**Figure 4 cells-11-03242-f004:**
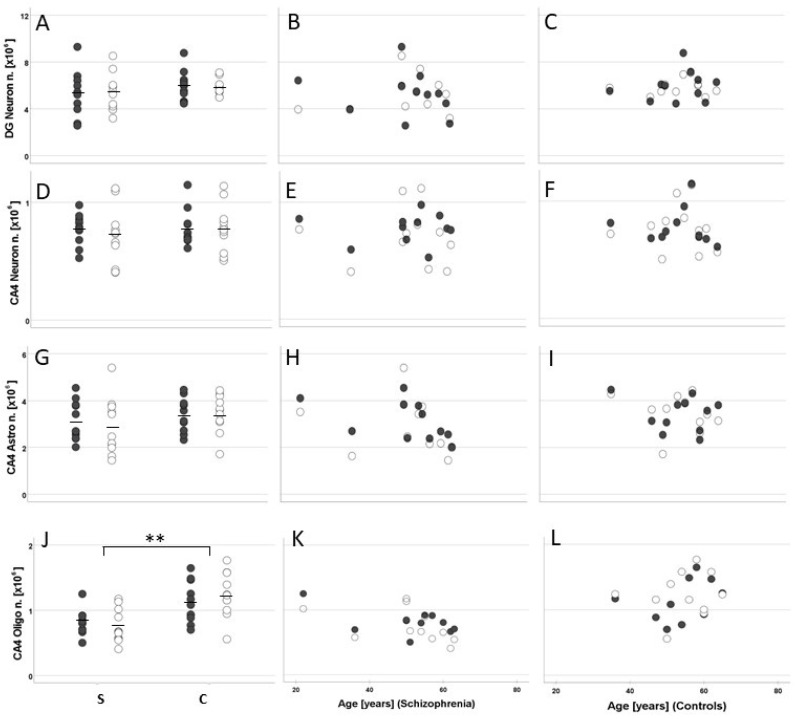
Mean and standard error of the mean (**A**,**D**,**G**,**J**) and individual values correlated with age (patients with schizophrenia [S]: (**B**,**E**,**H**,**K**); controls [C]: (**C**,**F**,**I**,**L**) of the total number of neurons in the hippocampal granule cell layer (the dentate gyrus; DG Neuron n.; (**A**–**C**)) and cornu ammonis 4 (CA4 Neuron n.; (**D**–**F**)) and of the number of astrocytes (CA4 Astro n.; (**G**–**I**)), and oligodendrocytes (CA4 Oligo n.; (**J**–**L**)) in postmortem brains from men with schizophrenia and age- and sex-matched controls. Closed dots represent data from left hemispheres, and open dots, data from right hemispheres. ** *p* < 0.005.

**Figure 5 cells-11-03242-f005:**
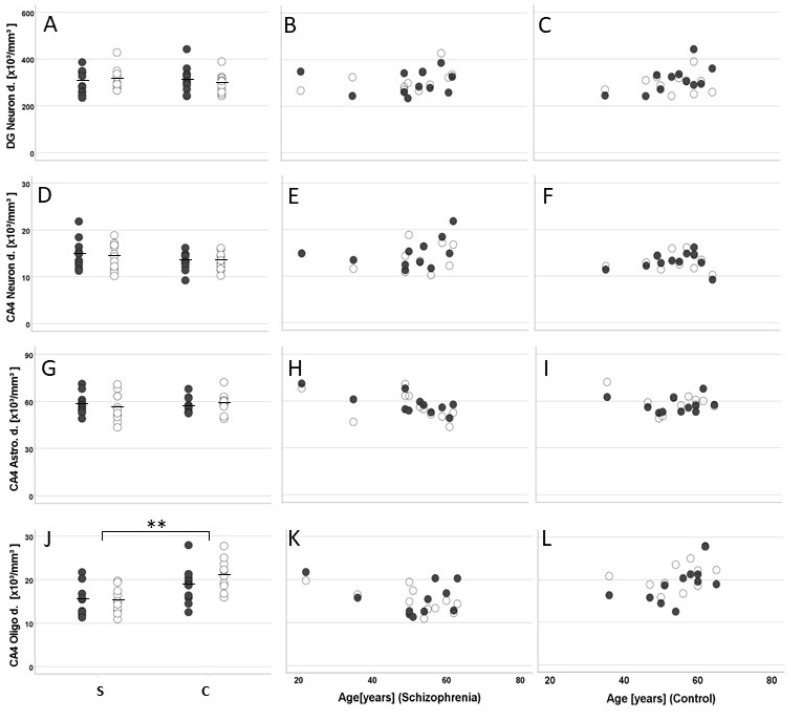
Mean and standard error of the mean (**A**,**D**,**G**,**J**) and individual values correlated with age (patients with schizophrenia [S]: (**B**,**E**,**H**,**K**); controls [C]: (**C**,**F**,**I**,**L**)) of the density of neurons (in 10^3^/mm^3^) in the hippocampal granule cell layer (the dentate gyrus; DG Neuron d.; (**A**–**C**)) and cornu ammonis 4 (CA4 Neuron d.; (**D**–**F**)) and of the density of astrocytes (CA4 Astro d.; (**G**–**I**)) and oligodendrocytes (CA4 Oligo d.; (**J**–**L**)) in postmortem brains from men with schizophrenia and age- and sex-matched controls. Closed dots represent data from left hemispheres, and open dots, data from right hemispheres. ** *p* < 0.005.

**Figure 6 cells-11-03242-f006:**
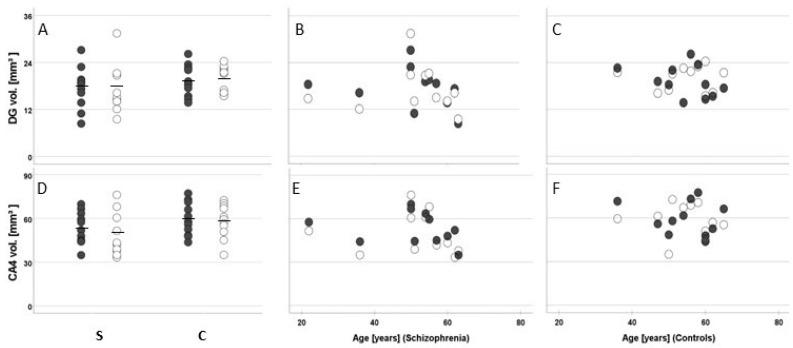
Mean and standard error of the mean (**A**,**D**) and individual values correlated with age (patients with schizophrenia [S]: (**B**,**E**); controls [C]: (**C**,**F**) of the volume (in mm^3^) of the hippocampal granule cell layer (the dentate gyrus; (**A**–**C**)) and cornu ammonis 4 subfield (CA4; (**D**–**F**)) in postmortem brains from men with schizophrenia and age- and sex-matched controls. Closed dots represent data from the left hemispheres, and open dots, data from right hemispheres.

**Table 1 cells-11-03242-t001:** Clinical characteristics of study participants.

No.	A	O	Cause of Death	PMI	Fix	Diagnosis	
	[y]	[y]		[h]	[d]	DSM-IV	ICD-10
S1	22	19	Suicide by jumping from high building	88	130	295.30	F20.00
S2	36	28	Suicide by strangulation	<72	115	295.30	F20.00
S4	50	17	Peritonitis	<24	203	295.30	F20.00
S5	50	22	Suicide by strangulation	18	170	295.30	F20.00
S6	51	17	Septicemia	33	127	295.60	F20.50
S7	54	20	Septicemia	27	250	295.60	F20.50
S8	55	22	Right-sided heart failure	25	84	295.30	F20.00
S9	57	37	Septicemia	76	163	295.30	F20.00
S10	60	24	Pulmonary embolism	<48	311	295.30	F20.01
S11	62	19	Aspiration	7	171	295.30	F20.00
S12	63	22	Acute myocardial infarct	15	338	295.60	F20.50
C2	36	-	Gunshot	24	143	-	-
C3	47	-	Acute myocardial infarct	<24	133	-	-
C5	50	-	Avalanche accident	23	498	-	-
C6	51	-	Septicemia	7	285	-	-
C7	54	-	Acute myocardial infarct	18	168	-	-
C8	56	-	Acute myocardial infarct	60	3570	-	-
C9	58	-	Acute myocardial infarct	28	126	-	-
C10	60	-	Gastrointestinal hemorrhage	18	101	-	-
C11	60	-	Gastrointestinal hemorrhage	27	302	-	-
C12	62	-	Acute myocardial infarct	<24	3696	-	-
C13	65	-	Bronchopneumonia	6	2289	-	-

A, age at death; C, matched control; Fix, fixation time; O, age at disease onset; PMI, postmortem interval (i.e., the time between death and autopsy); S, patient with schizophrenia.

**Table 2 cells-11-03242-t002:** Details of the stereological counting procedures.

	DG Neurons	CA4 Neurons	CA4 Astrocytes	CA4 Oligodendrocytes
Objective used	40	40	40	40
B [µm^2^]	625	1225	1225	1225
H [µm]	15	20	20	20
D [µm]	360	200	200	200
∑ CS	449	1131	377	1131
∑ Q-	321	358	506	472
CE_pred._ [n]	0.056	0.053	0.044	0.046
∑ P (Cavalieri)	114	1070	1070	1070

B, base of unbiased virtual counting spaces; CA4, cornu ammonis 4; CEpred [n], mean predicted coefficient of error for total cell numbers cells estimated with the prediction method of Schmitz [28] and Schmitz and Hof [29]; D, distance between the unbiased virtual counting spaces in mutually orthogonal directions x and y; Σ CS, mean sum of unbiased virtual counting spaces in one hemisphere; Σ Q-, mean number of neurons, astrocytes, and oligodendrocytes in one hemisphere (astrocytes were counted in every third counting space only); DG, dentate gyrus; H, height of unbiased virtual counting spaces; Σ P (Cavalieri), mean number of counted points (volume estimates).

## Data Availability

The data presented in this study are available on request from the corresponding author. The data are not publicly available for ethical reasons.

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
