# Peer review of "Decreased Oligodendrocyte Number in Hippocampal Subfield CA4 in Schizophrenia: A Replication Study"

_cells, 2022, doi:10.3390/cells11203242_

Round 1

Reviewer 1 Report

This manuscript by Andrea et al, investigated neuron, oligodendrocyte and astrocyte numbers of independent samples by utilizing a stereologic approach. The results indicated decreased oligodendrocytes bilaterally in CA4 in schizophrenia, suggesting oligodendrocyte and myelin deficits in schizophrenia brain. This is an interesting topic and the methods are generally sound. My major concern is that how well to identify oligodendrocytes, astrocyte and neurons by purely using histological staining? Is it possible there are a number of errors in identifying different cell types? Therefore, it is important to verify the accuracy of this approach by using a more specific labeling method. For example, how many OLs can be seen by using immunostaining for CC1, and how well this number can match with the HE staining result? A comparison between specific labeling and regular staining would be much more helpful and informative for the audience.  

Reviewer 2 Report

The manuscript is well written and the presented results confirm the stated hypothesis. 

Only a few minor issues need to be corrected.

Bottom of page 6 – repeated description for Figure 1

“Compared to controls, the schizophrenia patients showed a significantly reduced mean total number of oligodendrocytes (small dark cells) in the hippocampal subfield CA4 (-28% in the left hemisphere and -39% in the right hemisphere; F (1, 18) = 13.36, p = 0.002, Figure 3J).” Figure 3J should be 4J

It will be also nice to know how the exclusion criteria correlate to the cause of death in some cases. How it corresponds to the cases where the cause of death is listed as suicide (possibility of head trauma or substance abuse) and avalanche accident (possibility of head trauma).

Round 2

Reviewer 1 Report

The authors have addressed my concerns.